# The Effect of Age, Stage of the Annual Production Cycle and Pregnancy-Rank on the Relationship between Liveweight and Body Condition Score in Extensively Managed Romney Ewes

**DOI:** 10.3390/ani10050784

**Published:** 2020-04-30

**Authors:** Jimmy Semakula, Rene Anne Corner-Thomas, Stephen Todd Morris, Hugh Thomas Blair, Paul Richard Kenyon

**Affiliations:** School of Agriculture and Environment, Massey University, Private Bag 11222, Palmerston North 4410, New Zealand; R.Corner@massey.ac.nz (R.A.C.-T.); S.T.Morris@massey.ac.nz (S.T.M.); H.Blair@massey.ac.nz (H.T.B.); P.R.Kenyon@massey.ac.nz (P.R.K.)

**Keywords:** ewe, liveweight, liveweight change, body condition

## Abstract

**Simple Summary:**

Liveweight and body condition score (BCS)) are related, indicating that it may be possible to predict one from the other. The magnitude of this relationship can be altered by animal and environmental factors. The aim of the present study was to determine the nature of the association between liveweight and BCS over time, and the effect of the interaction between the stage of the annual cycle and the age of the ewe, using individual animal records. The association between liveweight and BCS was found to be linear and was affected by the interaction between ewe age, stage of the annual cycle and pregnancy-rank of the ewe. The results highlight the substantial contribution of BCS to the differences in liveweight of the ewe. The findings suggest that when predicting BCS from the liveweight, consideration of these factors is required, and different prediction equations are needed.

**Abstract:**

This study determined the nature of the relationship between liveweight and body condition score (BCS) and assessed the influence of the stage of the annual cycle and pregnancy-rank on the relationship between liveweight and BCS in Romney ewes. Data were collected from the same ewes at different ages (8–18, 19–30, 31–42, 43–54, 55–66 and ≥67 months), stages of the annual cycle (pre-breeding, at pregnancy diagnosis, pre-lambing and weaning) and pregnancy-rank (non-pregnant, single or twin). Linear regression was determined as being sufficient to accurately describe the relationship between liveweight and BCS. Across all data, a one-unit change in BCS was associated with 6.2 ± 0.05 kg liveweight, however, this differed by stage of the cycle, pregnancy-rank and ewe age (*p* <0.05). The average liveweight per unit change in body condition score increased with the age of the ewe and was greatest at weaning and lowest pre-lambing. Among pregnancy-ranks, the average liveweight per unit change was also greater during pregnancy diagnosis than pre-lambing and was greatest among single and lowest in non-pregnant ewes. The results support the hypothesis that the relationship between liveweight and BCS is affected by the interaction between stage of the annual cycle, pregnancy-rank and ewe age.

## 1. Introduction

Body condition score (BCS) is a subjective measure which provides an estimate of an animal’s soft tissue reserves (predominantly fat) and is used widely by farmers and researchers to determine the physiological state of an animal [1,2]. Body condition score was first developed for sheep (*Ovis aries*) by Jefferies [3] and was based on a 0–5 scale, using half units. Body condition score is assessed by the palpation of the lumbar vertebrae (spinous and transverse process) immediately caudal to the last rib and above the kidneys [3,4]. Body condition score can circumvent the shortcomings of liveweight (LW), which include the effect of gut fill, frame size, fleece weight and physiological state [4,5,6]. Body condition score can be easily learned and is cost-effective and requires no specialized equipment [4]. In addition, it has been suggested that BCS could be used to provide proper feeding management of a grazing flock throughout the year, detect subtle changes in a condition not noticeable by visual inspection, allow farmers to be more aware of major losses in condition and be used to follow changes in nutrition [3]. Body condition score is thus considered a useful way for farmers to monitor the condition of their flock and to estimate the required plane of nutritional allowance [4]. 

Despite the advantages of using BCS over liveweight to better manage flocks, it is uncommon for producers/farmers to regularly and objectively do so. A survey of sheep producers in Australia indicated that although 96% of respondents said they monitored the body condition of their sheep, only 7% conducted hands-on BCS assessment to estimate the energy requirements of their sheep [7]. In New Zealand, Corner-Thomas et al. [8] reported that the proportion of farmers using BCS as a management tool was 40%. The combination of these findings indicates that there is a sizable number of farmers not using BCS, especially in countries with large flocks. Besier and Hopkins [9] reported that farmers rely on a visual inspection method, that has been demonstrated to be very inaccurate, or prefer to use liveweight measures only. The reasons for low BCS uptake among farmers include: (1) Body condition score being subjective, depending on the judgement of the assessor; (2) it is labor-intensive and (3) requires training of the assessors, who should regularly undergo recalibration [4]. Strategies to increase the adoption and use of BCS among farmers and the reliability of measures included; promotional farmers’ training workshops and regular assessor recalibration [4]. However, given the apparently low rate of farmer use, these strategies appear not to have yielded the desired outcome, presumably because they do not directly address how to reduce the labor burden associated with hands-on BCS. Therefore, it is argued that consistent and accurate alternative methods to estimate body condition score of sheep that require less hands-on measurement would likely be advantageous and would improve uptake and use. Ideally, this prediction would be based on a management tool already utilized on farms, so that it reduces workload and would be quick and not subjective in nature.

Body condition score is correlated with liveweight and has been reported to have either a positive linear relationship [4,6] or a curvilinear relationship in ewes [10]. Factors such as breed, frame size, composition and patterns of fat distribution in the body [4,11] have been reported to affect the average change in liveweight associated with a one-unit change in BCS. The magnitude of the relationship between BCS and liveweight changes are related to the physiological status, age and breed of sheep [11,12]. Data on changes in either liveweight or BCS reflect changes in an animal’s body condition and can be used to inform decisions on appropriate feed allocation in a given physiological status and breed [13]. Therefore, assessment of the relationship between liveweight and BCS can be a valuable tool to maximize animal productivity and feed utilization [6,14]. The relationship between liveweight and BCS has generally been described by simple linear regression (based on R^2^), likely due to the simplest linear relationship appearing to be as strong as more complex models. However, using the coefficient of determination of a regression alone, as the criterion for goodness-of-fit, is not suitable to validate models because it does not provide information about the degree to which the predicted values diverge from true values [15,16]. Moreover, models should be robust in predicting other datasets. The majority of the previous studies have been based on fixed BCS ranges (mostly from 2.5 to 4.0), and it is unclear whether such a strong relationship would be observed in a wider range of BCS values (1 to 5). To date, no known attempts have been made to establish the true nature of the relationship between LW and BCS using a whole range of BCS values. It was hypothesized that the relationship between LW and BCS would be adequately described by linear regression. 

In cattle, the average liveweight change (i.e., gained or lost weight) associated with each BCS one-unit change is well associated with BCS [11,17]. Similar adjustment factors for sheep, however, have received less investigation. The current BCS adjustment factors in sheep have been generated from either relatively small-scale studies (*n* = 28, [6]; *n* = 156, [9]) or single time point observations (point specific) based on within-flock studies [4,11,12]. Ideally, the relationship between BCS and liveweight should be investigated using the same individuals over time. To the authors’ knowledge, no studies have conducted longitudinal studies for this purpose. Both conventional and modern weighing systems combined with individual electronic identification can now allow lifetime data to be collected more easily and quickly on large sheep flocks. Using this technology, combined with an individual BCS at a given point in their lifetime, therefore, can allow a specific stage of life BCS liveweight relationship to be developed. It was hypothesized that the relationship between liveweight and BCS would be modified by the stage of the annual cycle and pregnancy-rank over time. Therefore, this study had three objectives: using Romney ewes (i) to determine the nature of the relationship between liveweight and BCS, using both coefficient of determination and prediction error; (ii) to quantify the average liveweight change associated with each incremental change in BCS on a scale from 1 to 5, with 0.5-point intervals; and (iii) to determine if the association differed by stage of the annual cycle, pregnancy-rank, and over time. 

## 2. Materials and Methods 

### 2.1. Farms and Animals

The current study utilized datasets from a database collected between 2011 and 2015. Data was collected as part of normal routine farm management from two commercial New Zealand sheep farms in which all ewes were bred as ewe-lambs at approximately eight months of age at breeding. Farm A was located in the Waikato region of New Zealand and consisted of Romney ewes. Two cohorts of ewes from Farm (A) were included in this study: 2010-born (*n* = 3469) and 2011-born (*n* = 4572). Farm (B) was located in the Wairarapa region of New Zealand, with Romney ewes that were born in 2011 (*n* = 3760). The number of ewes monitored on each farm fluctuated by stage of the annual cycle. This was influenced by each farm owner’s decision to keep or dispose of (cull) ewes or failure to collect data during any period. Farm (A) did not collect liveweight and BCS data during the pre-lambing period on two occasions.

All ewes were weighed to the nearest 0.2 kg using static digital weighing scales (model XR5000, Tru-Test Group, Auckland, New Zealand) and were body condition scored (BCS) at the same time. Body condition score was assessed by palpating the soft tissue over the lumbar region on a 1–5 scale (1 = emaciated, 5 = obese) assessed to the nearest 0.5 unit [3,4]. BCS was assessed immediately prior to breeding (two to three weeks before the start of mating), at pregnancy diagnosis (approximately 80 days after start of mating), pre-lambing (within three weeks before the start of lambing) and at weaning (approximately 100 days after start of lambing). Body condition was measured over 6 years, beginning at approximately 8 months of age (age groups: 8–18, 19–30, 31–42, 43–54, 55–66 and ≥67 months). Body condition score was determined by two experienced assessors (one for the first 6 years (2011–2016) and one for the final year (2016)). The timing of measurements and the number of animals measured are summarized in Table A1 and Table A2. Additional information collected included: farm, year of observation, pregnancy-rank and age. The pregnancy-rank of the ewes was determined using transabdominal ultrasound conducted by a commercial operator (non-pregnant, 0; single foetus, 1; twin, 2). Due to the extensive nature of this study, data on individual nutritional, breeding and lambing dates could not be collected. 

### 2.2. Data Management

Liveweight and BCS data were first exported to Microsoft excel version 2010 for pre-processing, including cleaning, merging and validation. Data were then exported to the R statistical program version 3.3.4 [18] for further management. A total of 128,753 records from 11,798 ewes were collected between 2011 and 2016 (Table A1 and Table A2). Records were removed from the analysis dataset that had no identification (*n* = 15) or that had liveweight for the calibration weights (test weights) recorded (*n* = 9), so were removed from the analysis. The independent variables included: age group, determined by number of months at the time of breeding time within a 12-month period (i.e., 8–18, 19–30, 31–42, 43–54, 55–66 and ≥67 months); stage of annual cycle (pre-breeding, at pregnancy diagnosis, pre-lambing and weaning); and pregnancy-rank (non-pregnant, 0; single foetus, 1; twin, 2). In both farms, triplets (*n* = 67) were not considered due to fewer numbers and high variability in both liveweight and BCS compared to their contemporaries. A variable labelled FarmYear was generated to account for the different birth years as well as the farm of origin. When liveweight was considered as the dependent variable, BCS was considered its covariate and vice versa.

### 2.3. Statistical Analyses

All analyses were conducted in R program version 3.4.4 [18]. Pearson’s correlation between BCS and liveweight was estimated across all data and within each age group and stage of the annual cycle. Correlation coefficients were also estimated for each age group with adjustment for stage of the annual cycle and pregnancy-rank (for measurements made at pregnancy diagnosis and pre-lambing). Any significant differences between correlation coefficients were determined based on Fisher’s r-to-z transformation. 

#### 2.3.1. Model Development and Selection (Nature of Association between Liveweight and BCS) 

To determine the true nature of the relationship between liveweight and BCS, linear (LM), second-order polynomial/quadratic (QUAD), Box-Cox and square root (SQRT) transformation regressions were compared. The best lambda (with greatest likelihood) for Box-Cox transformation was 0.67. Table 1 gives the formulae by which the models, their goodness of fit (R^2^) and error metric (mean absolute error and percent error) were defined [19,20,21]. For this comparison, the percent error and the goodness-of-fit were based on the testing dataset. The models were adjusted for the effects of stage of the annual cycle, age group and FarmYear. The models were examined for normality of the residuals, and heteroscedasticity and outliers were examined using residual plots. In addition, Cooks distances were calculated for each model to assess the existence of outliers that may have influenced the coefficients of the models. The leverage plots were used to detect data points with an unusually high influence [22]. Outliers highlighted on the diagnostic plots were investigated and corrected if identified as a simple typing error or removed. The resulting dataset was then reanalyzed to determine its influence. In total, six of the 128,753 liveweight data points were removed. 

#### 2.3.2. Final Model Fitting (Factors Affecting the Relationship between LW and BCS)

The best linear model for final data fitting was selected by comparing two parameter estimation methods (a generalized least square vs. linear mixed-effects model). The linear mixed-effects model (LMM) was selected for fitting the model, as it had the smallest likelihood value and Akaike’s information criterion (AIC) values (*p* <0.001). To quantify the relationship between liveweight and BCS and the factors associated with this relationship, the final analysis was based on the minimal LMM model (with minimum Akaike’s information criterion, AIC value retained during simplification) incorporating all significant effects using the nlme package [23]. Three separate liveweight estimating models were constructed. The first model included body condition score (BCS) as a covariate, age group (A) and stage of the annual cycle (T) as explanatory variables (Appendix A, Model A1). To determine the impact of pregnancy-rank, two additional models (one for measurements at pregnancy diagnosis and another for pre-lambing) were constructed, each of the models taking a similar form (Appendix A, Model A2). In both models, BCS was treated as a covariate, age group and pregnancy-rank (P) as explanatory variables. To test whether BCS effects on liveweight were modified by age group, stage of the annual cycle and pregnancy-rank, the models included up to three-way interactions (BCS × A × T or BCS × A × P). A similar approach was used when assessing the effect of all other factors on BCS. FarmYear and individual ewe electronic identification number (EID) were included as random variables. Variance functions, to account for heteroscedasticity, and an autoregressive temporal correlation structure, to account for the temporal dependency of the nearby stage of the annual cycle, were also included. The differences between intercepts and slopes (beta coefficients) in the model were compared using Tukey’s pairwise contrasts on the final model using the multcomp package [24]. Statistical significance from the model using ANOVA type III is reported. To estimate the least squares means for BCS, the models above were refitted using BCS as the dependent variable and LW considered its covariate.

All models were constructed, fitted and cross-validated using machine learning algorithms, implemented in four steps. The steps included: (i) Data partitioning, (ii) resampling, (iii) model training and (iv) validation. Data partitioning involved dividing the initial dataset (with stratification preserving the class proportions) into training and testing datasets in a ratio of 3:1, with replacement. Resampling involved using bootstrapping and aggregation procedures [25,26] to select 10 subsamples from the training set and repeating the resampling five times. Model training involved fitting of the linear regression using the training dataset subsamples (10) from which, nine were used for computing the parameters (i.e., β) while the remaining one part was used for error estimation (ε). Finally, all parameters were used to determine the final value (estimate). Model cross-validation involved using the trained model to predict BCS in the testing dataset.

## 3. Results

A total of 128,753 ewe records were included in the analysis (Table A1 and Table A2). The number of records (*n*) decreased with ewe age. The majority of ewes were diagnosed as pregnant (93.3%, *n* = 32,764) with more ewes carrying twin foetus (56.9%, *n* = 19,987) compared to single (36.9%, *n* = 12,777) (Table A2). Body condition scores of 3.0 (41.6%, *n* = 56,381) and 2.5 (39.4%, *n* = 53,470) formed the bulk of the records while 1.0 (0.0%, *n* = 19) and 5.0 (0.0%, n = 6) were the least frequent (Table 2). The overall mean liveweight of ewes in this study was 54.2 kg (SD = 9.3 kg), and BCS was 2.81 (SD = 0.42). There was relatively high variability in liveweight for each BCS (mean CV = 15%, Table 2). 

### 3.1. Nature of Association between Liveweight and BCS 

The models were more stable at BCS from 2.5 to 3.5 (i.e., all model lines of best fit converged, as shown in Figure 1). All models had comparable statistical parameters (μ, SD) and were not significantly different from the observed data (Figure A1). Examination of the diagnostic plots for all four models that assessed the nature of the relationship between liveweight and BCS (in the initial construction) revealed that at the tails of the datasets were ‘hanging’ (not lying on the diagonal QQplot line; Figure A2) an indication that all the models were sensitive to bias at the extremes of the dataset. In addition, all models had relatively similar goodness-of-fit (R^2^ = 0.69) and Cook’s distances suggesting relatively similar robustness of these models to outlier effects (Table 3).

### 3.2. Effect of Age, Stage of Annual Cycle and Pregnancy-Rank on Ewe Liveweight and BCS

Age group, stage of the annual cycle and pregnancy-rank all affected ewe liveweight (*p* <0.05). As ewes increased in age, their liveweight increased (*p* <0.05) across all stages of the annual cycle, plateauing after 55–66 months (Figure A3). Ewes were heaviest (*p* <0.01) at pre-lambing in their last year of observation (≥67 months). Within age (except at 8–18 months), ewes were consistently heaviest (*p* <0.05) at pre-lambing. Among pregnancy-ranks, liveweight of ewes varied differently over time (*p* <0.05) with no clear pattern observed (Figure A4). There was, however, more variability in the liveweights of non-pregnant ewes than those bearing singles or twins. At pregnancy diagnosis, liveweight was lowest (*p* <0.05) in non-pregnant ewes in the first four age groups (8–18, 19–30, 31–42 and 43–54 months) compared to their contemporaries. Twin bearing ewes consistently had greater (*p* <0.05) liveweight than single or non-pregnant ewes across age up to the 43–54 months. Pre-lambing, liveweight was greater in twin than single bearing ewes (*p* <0.01) up to the 55–66 months. 

Body condition score was influenced (*p* <0.01) by age, stage of the annual cycle and pregnancy-rank (*p* >0.05). Body condition score decreased as the ewe increased in age (*p* <0.05) across all stages of the annual cycle plateauing after 55–66 months (Figure A5). However, when disaggregated by stage of the animal cycle and pregnancy-rank, BCS tended to decrease among the non-pregnant ewes at pregnancy diagnosis but with no clear pattern among other ranks (Figure A6). With the exception of age groups 31–42 and 55–66 months, across age, BCS was lowest (*p* <0.05) pre-lambing. Within the annual cycle and over time, the BCS of ewes showed no clear pattern of decline. Among pregnancy-ranks, BCS at pregnancy diagnosis was greater (*p* <0.05) in the first two years (8–18, 19–30 months) after which, it decreased remaining comparable (*p* >0.05) among pregnancy-ranks. Pre-lambing, BCS was greater in single than twin bearing ewes across age except for ≥67 months. 

### 3.3. Effect of Age, Stage of Annual Cycle and Pregnancy-Rank on the Relationship between Liveweight and BCS

Overall, the correlation between BCS and liveweight was 0.47, indicating that 21% (R^2^ = 0.21) of the variability in liveweight was explained by differences in BCS. When adjusted for age, stage of the annual cycle and pregnancy-rank, the overall correlation decreased slightly to 0.44 (R^2^ = 0.18). The correlation between BCS and liveweight was affected by both age of the ewe, stage of the annual cycle and pregnancy-rank (*p* <0.05). Overall, the correlation between liveweight and BCS varied from 0.02 pre-lambing to 0.69 at pregnancy diagnosis (Table 4 and Table 5). 

The strength of the association between BCS and liveweight differed significantly (*p* <0.05) across both age of ewe and stage of the annual cycle. Within the age group, the correlation between liveweight and BCS was relatively similar except at ≥67 months. Within the stage of the cycle, the correlation between liveweight and BCS was strongest at weaning and weakest at pre-lambing. Within pregnancy-rank, the correlation between liveweight and BCS varied from 0.02 (*p* >0.05) pre-lambing to 0.69 (*p* <0.01) at pregnancy diagnosis. There was no clear pattern in the strength of association among age groups and pregnancy-ranks (*p* >0.05).

Table 4 and Table 5 summarize the regression equations of the relationship between liveweight and BCS by age of ewe, stage of the cycle and pregnancy-rank. The regression intercepts, as well as the average change in liveweight per one-unit change in BCS (incremental liveweight change), were affected by all three factors (*p* <0.05).

The incremental liveweight change increased (*p* <0.001) as the ewes aged. The magnitude of the incremental liveweight change of ewes in the same age group (Table 4) was altered by the stage of the annual cycle (*p* <0.001). The increase in the average incremental liveweight change varied from 2.3 kg for younger ewes pre-lambing (8–18 month) to 9.5 kg for the older ewes at weaning (≥67 months).

Within the stage of the annual cycle, the incremental liveweight change was lowest (*p* <0.01) at 8–18 months but the maximum change varied by stage of the annual cycle, for example at ≥67 months for pre-breeding and at pregnancy diagnosis, 43–54 for pre-lambing and 55–66 for weaning. Weaning was associated with the greatest incremental liveweight change (5.6 to 9.5 kg) while pre-lambing was associated with the lowest (2.3 to 5.9 kg) (*p* <0.05). 

Among pregnancy-ranks, the increase in incremental liveweight change was greater (*p* <0.01) at pregnancy diagnosis (4.3 to 13.6 kg) compared to pre-lambing and increased with the age of the ewe. Pre-lambing, the increase in incremental liveweight change had no clear pattern. Generally, incremental liveweight change for similar age groups appears to have varied randomly regardless of pregnancy-rank (Table 5). At pregnancy diagnosis, the incremental liveweight change was greater (*p* <0.05) in single and twin bearing ewes than non-pregnant ewes at all age groups except at 8–18 and ≥67 months. The incremental liveweight change was also comparable (*p* >0.05) for both single and twin bearing ewes except at 19–30 months. Pre-lambing, the incremental liveweight change was unexpectedly low (0.4 to 3.8 kg) and varied with no clear pattern among pregnancy-ranks as the ewe aged. 

## 4. Discussion

This study aimed to determine the nature of the association between liveweight and BCS and to quantify the average liveweight change associated with each incremental change in ewe BCS as measured on a 1 to 5 scale with 0.5-point intervals. In addition, the extent to which this association differed by stage of the annual cycle and pregnancy-rank and ewe age, in extensively managed Romney ewes was investigated. It was hypothesized that the relationship between BCS and liveweight was best described using a linear regression and would vary based on age group, stage of cycle and pregnancy-rank. 

In the present study, the linear regression was considered sufficient to describe the relationship between liveweight and body condition score. This was not surprising as the majority of previous studies have reported a linear relationship between liveweight and BCS [4]. In addition, transforming data would add unneeded complexity to the model [27]. The percent error for all the four models (LM, QUAD, Box-Cox and SQRT) was within acceptable range (i.e., <10%), for veterinary purposes [28] and prediction models [29]. The findings show, therefore, that liveweight and BCS vary together in a linear manner, and this relationship ship can be predictable using simple linear regression.

Liveweight increased with the ewe age and began to plateau at 43 months of age. Ewe liveweight increased with frame size as an animal aged, until its mature size was achieved [30]. In temperate (European) sheep breeds, this has been reported to occur between 25–50 months of age [31,32]. The present findings are in agreement with other authors who reported a liveweight increase with age, plateauing after 33 months of age in Romney ewes [33,34]. In three thin-tailed breeds of indigenous Turkish sheep, liveweight increased with age [12], although that study did not have age groups below two years to demonstrate the overall trend.

Within the age group, ewe liveweight was highest at pre-lambing. During late pregnancy, the conceptus weight influences total ewe liveweight, as single, twin and triplets near term can add 5–8 kg, 12–17 kg and 17–21 kg, respectively [35,36]. Thus, it was perhaps not surprising that ewe liveweight was heaviest pre-lambing and tended to increase with pregnancy-rank. The observed low liveweight among the non-pregnant ewes in the first three years, particularly during at pregnancy diagnosis may be explained by the fact that lighter ewes are less likely to conceive. 

There was a general decline in BCS with the age of the ewe, which began to plateau from 55 months of age across the stages of the annual cycle. This finding is in agreement with a declining trend for BCS with age at breeding in Merino and Corriedale ewes [37]. However, it has been reported that thin-tailed breeds of indigenous Turkish sheep had greater BCS scores pre-breeding but lower scores at lambing and weaning across age groups [12]. The results of the current study contrast with others who reported greater condition scores as a ewe aged across all stages of the annual cycle in mature mixed sheep breeds and crossbreds [11]. Breed differences and nutritional conditions may explain the differences observed between studies. However, we did not collect data on nutritional status due to the extensive nature of the study and given that was conducted over multiple seasons and years. A declining BCS over time indicates that ewes used their body reserves to meet their nutritional demands, thus, suggesting that at times ewes in this study were likely not being fed to meet their theoretical nutritional requirements, particularly in lactation. The change in trend of BCS when data were disaggregated by pregnancy-rank highlights a potential interaction between factors affecting BCS in sheep. The ewes found to be non-pregnant during pregnancy diagnosis in the first two age groups (8–18 and 19–30 months) were also lighter. The finding therefore agrees with previous studies [38,39,40]. 

The correlation between BCS and liveweight was weak to moderate based on a scale of 0 to 1.0 [41,42], ranging between 0.18 and 0.67 across ages, stages of the annual cycle and pregnancy-rank. By adjusting for age and stage of the annual cycle, these results suggest that 6% to 45% of the variability in liveweight was explained by differences in BCS and vice versa. These values are lower than those previously reported by others 0.60 to 0.82 [12] from data of 156 ewes and 0.81 from data of 28 mixed aged Romney ewes [6]. They are comparable, however, to those reported for a study with multiple breeds (0.36 to 0.63) and stages of the annual cycle (0.42 to 0.62) [11]. The between studies difference in correlation strength may be explained by variation in sample sizes, breed, stage of the annual cycle and study design. The weaker correlation between BCS and liveweight observed at pre-lambing could be attributed to the difficulty (data can be more variable) to body condition score heavily pregnant ewes [4,43].

In this study, a linear relationship between liveweight and body condition score was demonstrated. This relationship was affected by ewe age, stage of the annual cycle and pregnancy-rank. These results are in agreement with previous findings showing significant age and stage of the annual cycle effects [4,6,11]. A linear relationship suggests that, for a given breed type, a single incremental liveweight change across the entire BCS range can be applied. The incremental liveweight change increased with the age of the ewe and varied across the stage of the annual cycle being numerically lowest at 8–18 months and greatest at ≥67 months. Thus, as a ewe ages, a greater liveweight change is required to alter BCS by one unit, which translates into greater energy requirements in order to make the change [6,44].

The relationship between BCS and liveweight also varied by stage of the annual cycle. Overall, the liveweight change required to cause a one-unit change in BCS was greatest at weaning and lowest at pre-lambing. It is not clear why the regression coefficients of liveweight on BCS at pre-lambing were consistently low. It may have been because the conceptus and uterine mass was not accounted for, which is likely to have confounded the true liveweight change associated with a unit change in BCS. The conceptus mass has an influence on total ewe liveweight from mid-pregnancy [45,46] which coincides with pregnancy diagnosis. Additionally, it may have been due to the difficulty associated with body condition scoring of pregnant animals [4,43] as previously stated. Among mature ewes (≥43 months), the incremental liveweight change during mating/breeding was within the range reported for mixed-age Romney ewes [4,6], but were greater than Romney composite ewes [4,39,40]. 

Pregnancy-rank significantly affected the liveweight of ewes, their body condition scores (BCS) and eventually the relationship between liveweight and BCS. The effect of pregnancy-rank on liveweight was not surprising given that ewe liveweight was potentially confounded by fetal weight from mid to late pregnancy. The effect of pregnancy-rank on BCS is in agreement with earlier findings in Romney sheep [40], merino sheep [47], Cheviots [48,49] and in Scottish blackface ewes [50].

The finding that the incremental liveweight change was lower in non-pregnant ewes at pregnant diagnosis was not surprising as their energy demand would be expected to be lower than for pregnant ewes. The energy demand is greater for pregnant ewes and increases with the number of fetuses [51]. It is, however, not clear why the incremental liveweight change at pre-lambing varied randomly. The unexpectedly low incremental liveweight change among pregnancy-ranks at pre-lambing could have resulted from the confounding effect of the fully-grown fetal weight. 

## 5. Conclusions

In conclusion, the current study demonstrated that in a large population of ewes across a full range of BCS, liveweight and BCS were linearly related, and the relationship depended on the age of ewe, stage of the annual cycle and pregnancy-rank, therefore, supporting our hypothesis. The results indicate that large variability exists in BCS, and BCS contributes substantially to the differences in liveweight. The findings suggest that when predicting BCS from liveweight, consideration of these factors is required, and different prediction equations are needed. Adjustments for differences between BCS should consider age group, stage of the annual cycle and pregnancy-rank. The relationships found between liveweight and body condition score support the possibility of using liveweight as a proxy for body condition score. 

## Figures and Tables

**Figure 1 animals-10-00784-f001:**
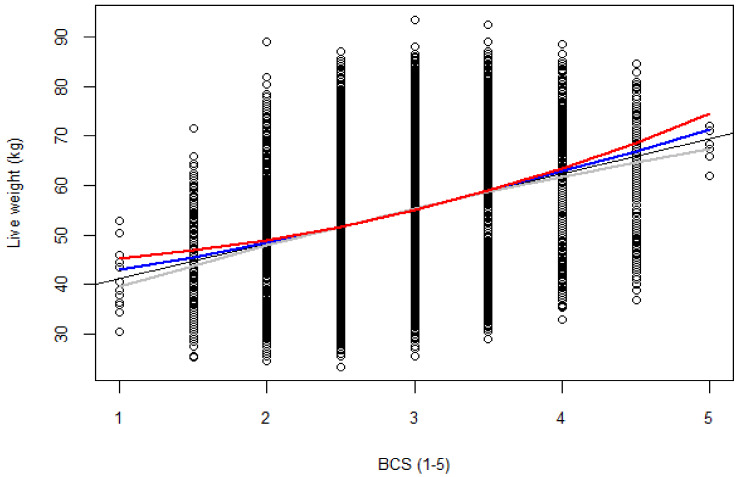
Ewe liveweight (kg) as a function of BCS (1–5). Line of best fit is given for (linear model (SLM), black color; quadratic transformation (QUAD), grey; Box-Cox transformation (Box-Cox), red; and square root transformation (SQRT), blue).

**Table 1 animals-10-00784-t001:** Formulae for liveweight estimation models (linear model (SLM), quadratic transformation (QUAD), Box-Cox transformation (Box-Cox) and square root transformation (SQRT)) using body condition score (BCS), adjusted R^2^, error metrics (mean absolute error, MAE; percent error, PE) and coefficient of variation of the liveweight (LW).

Model / Measure	Formula
Linear model (LM)	LW=α+BCS
Quadratic transformation (Quad)	$$\mathrm{LW}=\alpha +b(\mathrm{BCS})+c{\left(\mathrm{BCS}\right)}^{2}$$
Square root transformation (SQRT)	$${\mathrm{LW}}^{0.50}=\alpha +\mathrm{BCS}$$
Box_Cox transformation (Box_Cox)	$${\mathrm{LW}}^{0.67}=\alpha +\mathrm{BCS}$$
Adjusted R^2^ (Adj.R^2^)	$$\mathrm{Adj}.R^{2}=1-\frac{{\mathrm{MS}}_{\mathrm{RES}}}{{\mathrm{MS}}_{\mathrm{TOT}}}$$
Mean Absolute Error (MAE)	$$\mathrm{MAE}=\frac{1}{n}\sum_{j=1}^{n}(|y_{j}-{\hat{y}}_{j}|)$$
Mean Absolute Error Percent (PE)	$$\mathrm{PE}=\frac{1}{n}\sum_{j=1}^{n}(|\frac{y_{j}-{\hat{y}}_{j}}{y_{j}}|)\times 100$$
Coefficient of variation (CV)	$$\mathrm{CV}=100\sqrt{\frac{MS_{R}}{\mathrm{mean}}}$$

α indicates the intercept; *bc* indicates the regression coefficients; *y_j_* indicates the actual expected output; *ŷ_j_* indicates the model’s prediction; *n* indicates the sample size. MS_R_ indicates variation due to the model; MS_Tot_ indicates total variation.

**Table 2 animals-10-00784-t002:** The number of records (*n*), mean and standard deviation (SD), coefficient of variation (CV%) for liveweight across BCS.

BCS (Units)	Number of Records (*n*)	Liveweight (Kg)		
		Mean	SD	CV%
1.0	19	41.5	5.6	13.4
1.5	350	45.6	8.1	17.7
2.0	7735	49	8.1	16.6
2.5	53,470	51.6	8.6	16.7
3.0	56,381	55.8	8.8	15.7
3.5	15,051	59.4	9.7	16.4
4.0	2350	62.2	10.7	17.2
4.5	241	60.6	11.2	18.5
5.0	6	67.8	3.6	5.3

**Table 3 animals-10-00784-t003:** Mean absolute error (MAE) and percent error (PE), adjusted R^2^ and percentiles of Cook’s distance of the models (linear model (SLM), quadratic transformation (QUAD), Box-Cox transformation (Box-Cox) and square root transformation (SQRT)) for liveweight predictions on testing dataset.

Model	SLM	QUAD	SQRT	Box-Cox
Adjusted R^2^	0.69	0.69	0.69	0.69
MAE	4.12	4.11	0.28	0.74
*p*-value	***	***	***	***
Percentiles of PE				
75th	7.8%	7.7%	7.8%	7.8%
90th	7.8%	7.8%	7.8%	7.8%
95th	7.8%	7.8%	7.8%	7.9%
Percentiles of Cook’s distance				
75th	0.00001	0.00001	0.00001	0.00001
90th	0.00003	0.00002	0.00003	0.00003
95th	0.00005	0.00004	0.00004	0.00004

*** indicate significance at *p* < 0.001.

**Table 4 animals-10-00784-t004:** Intercepts (α), coefficients (β), correlation coefficient (r_xy_) and adjusted R^2^ for the regression of the liveweight with body condition score for each stage of the annual cycle (pre-breeding, at pregnancy diagnosis, pre-lambing, weaning) and ewe age (8–18 months, 19–30, 31–42, 43–54, 55–66 and ≥67).

Stage of Annual Cycle	Age Group	α (*SE*)	β (*SE*)	r_xy_	Adj. R^2^
Pre-breeding	8–18	33.2 (0.25)	2.8 (0.09) ^a^	0.43 ^bc^	0.15
	19–30	36.5 (0.29)	6.0 (0.10) ^d^	0.49 ^bc^	0.24
	31–42	36.9 (0.36)	7.1 (0.13) ^ef^	0.50 ^bc^	0.26
	43–54	39.5 (0.4)	6.9 (0.13) ^e^	0.48 ^bc^	0.28
	55–66	46.0 (0.39)	5.8 (0.14) ^d^	0.48 ^bc^	0.23
	≥67	37.6 (0.72)	8.4 (0.23) ^g^	0.58 ^c^	0.35
At pregnancy diagnosis	8-18	34.9 (0.25)	2.8 (0.09) ^a^	0.41 ^bc^	0.13
	19–30	35.6 (0.32)	5.0 (0.12) ^c^	0.34 ^b^	0.15
	31–42	38.3 (0.35)	5.9 (0.12) ^d^	0.49 ^bc^	0.26
	43–54	38.4 (0.41)	7.0 (0.14) ^ef^	0.45 ^bc^	0.21
	55-66	40.8 (0.49)	7.0 (0.17) ^ef^	0.45 ^bc^	0.23
	≥67	42.1 (0.72)	7.2 (0.22) ^ef^	0.56 ^c^	0.31
Pre-lambing	8–18	42.6 (0.34)	2.3 (0.12) ^a^	0.06 ^a^	0.24
	19–30	50.5 (0.38)	2.4 (0.14) ^a^	0.14 ^a^	0.06
	31–42	48.9 (0.54)	4.0 (0.19) ^b^	0.29 ^a^	0.10
	43–54	48.3 (0.44)	5.9 (0.16) ^d^	0.13 ^a^	0.21
	55–66	52.2 (0.61)	5.3 (0.21) ^cd^	0.13 ^a^	0.10
	≥67	57.2 (0.92)	4.8 (0.35) ^bcd^	0.32 ^ab^	0.07
Weaning	8–18	30.9 (0.25)	7.5 (0.09) ^f^	0.57 ^c^	0.45
	19–30	38.3 (0.27)	5.6 (0.09) ^d^	0.57 ^c^	0.28
	31–42	35.9 (0.34)	7.4 (0.11) ^ef^	0.58 ^c^	0.36
	43–54	36.1 (0.38)	8.3 (0.14) ^g^	0.62 ^c^	0.30
	55–66	34.8 (0.43)	9.5 (0.16) ^h^	0.62 ^c^	0.40
	≥67	39.8 (0.86)	7.5 (0.3) ^efg^	0.64 ^cd^	0.41

^a–n^ superscripts within a column indicate significant difference at *p* <0.05; *SE* denotes standard error.

**Table 5 animals-10-00784-t005:** Intercepts (α), coefficients (β), correlation coefficient (r_xy_) and adjusted R^2^ for the regression of the liveweight with body condition score value for each age (8–18 months, 19–30, 31–42, 43–54, 55–66 and ≥67) by pregnancy-rank (non-pregnant, single and twin bearer) and stage of the annual cycle (at pregnancy diagnosis and pre-lambing).

Pregnancy-Rank	Age Group (Months)	α (*SE*)	β (*SE*)	r_xy_	Adj. R^2^
*At pregnancy diagnosis*					
Non-pregnant	8–18	21.0 (1.17)	9.4 (0.41) ^d^	0.59 ^c^	0.06
Single		29.9 (0.54)	7.5 (0.19) ^b^	0.37 ^b^	0.05
Twin		29.3 (0.55)	7.9 (0.19 ^b^	0.44 ^ab^	0.17
Non-pregnant	19–30	23.4 (2.43)	8.3 (0.9) ^de^	0.69 ^cde^	0.15
Single		27.2 (0.56)	12.1 (0.2) ^h^	0.56 ^c^	0.16
Twin		31.6 (0.47)	7.2 (0.16) ^b^	0.41 ^b^	0.12
Non-pregnant	31–42	36.7 (3.08)	4.3 (1.14) ^a^	0.38 ^c^	0.43
Single		20.9 (0.62)	10.9 (0.29) ^ef^	0.53 ^bc^	0.31
Twin		26.8 (0.48)	8.9 (0.17) ^cd^	0.47 ^bc^	0.28
Non-pregnant	43–54	28.8 (2.43)	8.0 (0.87) ^c^	0.36 ^b^	0.47
Single		20.5 (0.75)	11.1 (0.26) ^ef^	0.52 ^b^	0.20
Twin		23.7 (0.51)	10.1 (0.18) ^ef^	0.48 ^b^	0.16
Non-pregnant	55–66	27.5 (2.51)	8.7 (0.88) ^ef^	0.53 ^b^	0.17
Single		19.1 (0.84)	11.6 (0.29) ^fg^	0.50 ^ab^	0.15
Twin		22.8 (0.55)	10.4 (0.19) ^e^	0.49 ^b^	0.15
Non-pregnant	≥67	30.4 (2.31)	7.8 (0.80) ^bc^	0.32 ^cd^	0.27
Single		18.4 (1.61)	11.8 (0.55) ^g^	0.47 ^c^	0.34
Twin		13.2 (0.85)	13.6 (0.29) ^hi^	0.52 ^c^	0.27
*Pre-lambing*					
Non-pregnant	8–18	-	-	-	-
Single		52.0 (0.73)	1.2 (0.26) ^c^	0.06 ^a^	0.02
Twin		45.9 (0.78)	3.8 (0.29) ^e^	0.04 ^a^	0.01
Non-pregnant	19–30	-	-	-	-
Single		51.6 (0.74)	1.6 (0.46) ^cd^	0.05 ^b^	0.01
Twin		52.7 (0.63)	1.3 (0.43) ^cd^	0.06 ^b^	0.12
Non-pregnant	31–42	-	-	-	-
Single		52.2 (0.8)	1.6 (0.29) ^cd^	0.04 ^bc^	0.02
Twin		52.7 (0.64)	1.5 (0.23) ^cd^	0.06 ^b^	0.03
Non-pregnant	43–54	-	-	-	-
Single		50.5 (0.96)	2.2 (0.35) ^cd^	0.11 ^ab^	0.01
Twin		51.7 (0.66)	2.0 (0.24) ^cd^	0.02 ^b^	0.10
Non-pregnant	55–66	-	-	-	-
Single		54.4 (1.02)	0.9 (0.36) ^abc^	0.06 ^ab^	0.08
Twin		50.1 (0.71)	2.6 (0.26) ^de^	0.05 ^ab^	0.04
Non-pregnant	≥67	-	-	-	-
Single		50.3 (1.94)	2.4 (0.69) ^cde^	0.15 ^c^	0.02
Twin		58.8 (1.03)	0.4 (0.37) ^ab^	0.02 ^b^	0.01

- indicates data not collected; ^a–j^ Different superscripts within a column and stage of the annual cycle indicates differences at *p* <0.05. *SE* denotes standard error.

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
