# Peer review of "The Effect of Age, Stage of the Annual Production Cycle and Pregnancy-Rank on the Relationship between Liveweight and Body Condition Score in Extensively Managed Romney Ewes"

_animals, 2020, doi:10.3390/ani10050784_

Round 1

Reviewer 1 Report

The topic is actual, concerning frequently discussed tool of flock management. Authors have experience in this field which is demonstrated by references on their previous research. The purpose of the trial is well supported by literature overview in Introduction. The authors had available a big set of animals which makes their dataset robust for further evaluation. They also used elaborated statistical procedures. Anyway, number of procedures described together is sometimes difficult for understanding. Consider attaching model equation with statistical description in supplementary file in appendix section. This possibility could help to the reader for better orientation in particular statistical procedures when needed.
Tables with linear regression (Table 4 and Table 5): Estimation of significant differences among intercepts and coefficients across categories is redundant for me personally. Instead of that, I would appreciate information about significance of individual regression and correlation coefficients.  This would be more informative for prediction you mentioned. Additionally, significant differences across of age group, stage of annual cycle, or pregnancy diagnosis can be identified in tables in supplementary file sufficiently.   

Author Response

Response to reviewers’ comments

We thank the reviewer for all their comments. We have reviewed the reviewer’s comments. Each comment (point) is below in black followed by our response in red. In addition, we have made corrections where necessary and given more detail to improve the clarity of the manuscript.

Reviewer 1 Comments

Point 1: The topic is actual, concerning frequently discussed tool of flock management. Authors have experience in this field which is demonstrated by references on their previous research. The purpose of the trial is well supported by literature overview in Introduction. The authors had available a big set of animals which makes their dataset robust for further evaluation. They also used elaborated statistical procedures.

Response to point 1: We thank the reviewer for the overall comment.

Point 2: Anyway, number of procedures described together is sometimes difficult for understanding. Consider attaching model equation with statistical description in supplementary file in appendix section. This possibility could help to the reader for better orientation in particular statistical procedures when needed.

Response to point 2: change made (see pages 5, 23 and 24)

We have now attached model equations (Model A1, Model A2) with statistical description in supplementary material (lines 601 to 623 and 634 to 652). In addition, the model equations are now referred to in the methods section (see lines 189 to 190 and 191 to 192)

Point 3: Tables with linear regression (Table 4 and Table 5): Estimation of significant differences among intercepts and coefficients across categories is redundant for me personally. Instead of that, I would appreciate information about significance of individual regression and correlation coefficients.  This would be more informative for prediction you mentioned. Additionally, significant differences across of age group, stage of annual cycle, or pregnancy diagnosis can be identified in tables in supplementary file sufficiently.   

Response to point 3: change made (see pages 5, 8 and 9)

We now only highlight the significance of individual regression and correlation coefficients in the tables 4 and 5.  We have removed the letters of significant difference that were previously on the intercepts in these tables. Also, in regard to the test associated with these tables, reference is made to lines 200 to 201.

Reviewer 2 Report

It is a great job with a large number of animals followed for a long time and continued by good statistical analysis.

I only point out a few small changes.

Minor changes:

-Line 67: Please, define the acronym LW since it is the first time it appears in the text

-Line 120: I think it will be better understood if it is changed ; by :

...included:  age...

-Line 122: Likewise change , by ; 

...months); stage...

-Line 123: insert ; 

...weaning); and...

-Line 190: For better tex uniformity change 32764 by 32,762

-Line 204: Write Cook´s with a capital letter

-Line 207: Review the lines of relationship types. I think that the acronyms that are assigned to the colors in the box inside the figure do not match with those of the figure footnote

-Line 207: Please define all the acronyms of Figure 1 footnote so that they agree with those inside the table

-Line 210: In Table 3 footnote, not all the acronyms are defined. And write Cook´s with a capital letter

-Line: 278: Here type pregnancy with lower case

Author Response

Response to Reviewer 2 Comments

We thank the reviewer for all their comments. We have reviewed the reviewer’s comments. Each comment (point) is below in black followed by our response in red. In addition, we have made corrections where necessary and given more detail to improve the clarity of the manuscript.

Point 1:  It is a great job with a large number of animals followed for a long time and continued by good statistical analysis. I only point out a few small changes.

Response to point 1: We thank the reviewer for the overall comment.

Point 2:  -Line 67: Please, define the acronym LW since it is the first time it appears in the text

Response to point 2: change made (see line 43, page 1)

A definition of acronym LW has been given and now line reads “live weight (LW)”

Point 3: -Line 120: I think it will be better understood if it is changed ; by :

...included:  age...

Response to point 3: change made (see line 142, page 3).

We have replaced semicolon with full colon and now reads “included: age …”

Point 4: -Line 122: Likewise change , by ; 

...months); stage...

Response to point 4: change made (see line 144, page 3).

We have replaced comma with semicolon and now reads “... ≥67 months); stage …”

Point 5: -Line 123: insert ;  ...weaning); and...

Response to point 5: change made (see line 145, page 3).

We have replaced comma with semicolon and now reads “…weaning); and pregnancy-rank...

Point 6: -Line 190: For better tex uniformity change 32764 by 32,762

Response to point 6: change made (see line 218, page 5).

We have reformatted 32764 to 32,762.

Point 7: -Line 204: Write Cook´s with a capital letter

Response to point 7: change made (see line 233, pages 6).

We have edited all cook’s to Cook’s.

Point 8: -Line 207: Review the lines of relationship types. I think that the acronyms that are assigned to the colors in the box inside the figure do not match with those of the figure footnote

Response to point 8: changes made (see lines 237 to 239, page 7).

The acronyms were differing from the predefined one but have now been standardized and matched with line colour and footnotes. Linear, SLR and SLM all reconciled to one acronym (SLM) represented by black colour in the figure. Quadratic transformation, Quad and QUAD are all reconciled to one acronym (QUAD) represented by grey colour in the figure. Box_Cox transformation is denoted by Box_Cox represented by blue colour in the figure. Square root transformation is denoted by SQRT and represented by red colour in the figure.

Point 9: -Line 207: Please define all the acronyms of Figure 1 footnote so that they agree with those inside the table

Response to point 9: changes made (see lines 237 to 239, pages 7, 18 to 19).

All acronyms have now been defined and matched with figure legend and footnotes. SLM (linear model): black line, QUAD (Quadratic transformation): grey line, Box_Cox (Box_Cox transformation): blue line and SQRT (Square root transformation): red line.

Similarly, we have edited figure A2 (see page 18 to 19) label from LM to SLM both on the graph and footnote. Figure A2 has also been improved to show quartile ranges (dotted blue lines) for a simulated robust regression line (blue solid line).

Point 10: -Line 210: In Table 3 footnote, not all the acronyms are defined. And write Cook´s with a capital letter

Response to point 10: changes made (see line 240 to 242, page 7).

All acronyms have now been defined in the Table 3  title which now reads “Mean Absolute Error (MAE) and Percent Error (PE), Adjusted R2 and percentiles of Cook’s distance of the models (Linear model (SLM), Quadratic transformation (QUAD), Box_Cox transformation (Box_Cox) and Square root transformation (SQRT)) for live weight predictions on testing dataset”. In addition, … cook’s … has been corrected to … Cook’s …

Point 11: -Line: 278: Here type pregnancy with lower case

Response to point 11: change made (see line 328, page 10).

Pregnancy has been edited and now reads “At pregnancy diagnosis…”.

Reviewer 3 Report

The study uses an extensive data set and model development to relate live weight to body condition score changes. My two major concerns with the study are (1) the limited effective range in body condition scores despite the large overall data set, and (2) its biological relevance. Of the 128,753 ewe records, over 97% represented only 3 BCS scores (2.5, 3.0 and 3.5) on the 9 point scale (just the two scores 2.5 and 3.0 represent more than 85% of records). While I appreciate the constraints of identifying a wider range of scores in a commercial setting and not being a statistician, I am sure how a meaningful regression analysis can be conducted over such a narrow range. This would also explain why such low r-values were obtained from such a large data set.

As to the second concern, is there is really a need to predict BCS from live weight (line 377) or use live weight as a proxy for BCS (line 380/381) when BSC is a measure that can be more readily recorded than live weight? Not sure this represents valid conclusions. A stronger case needs to be made by the authors as to the need for these prediction models and their practical applications.

Apart from these major concerns, there a number of smaller issues that should to be addressed by authors.

Line 24; ‘Pregnancy diagnosis’ is an arbitrary measure dependent on management system, ‘mid-pregnancy’ should be used to more clearly define stage of production cycle.

Line 28; ‘Pregnancy-rank’ should be replaced with ‘pregnancy status’ to more accurately reflect what the authors are referring to. Also, with this many parturitions, were there no triplet birth? If so, the pregnancy status should be changed to ‘multiples’, unless these records were deleted from the data set.

Line 77; electronic ID and modern weigh system would not be ‘pre-requisites’ for a study to collect lifetime data (given it would help)

Line 93; table A1 and A2 refer to animal numbers not management

Line 110; there is no table A4

Figures A4/5; what is the relevance of measuring significance of differences in live weight across age groups and production stages, wouldn’t it be more meaningful if it would be corrected for these variables (of course a ewe is heavier when late pregnant and then at breeding)

Line 224; superscripts in Figure A4 would suggest that live weight of single and twin bearing ewes are not different at 55-66 and >67 months as is stated in the text

Line 235; BCS according to the superscripts in Figure A6 indicate that differences in BCS between age groups was not significant, also title for Figure A6 should say ‘body condition score’, rather than ‘live weight’

Line 265; define incremental live weight change here, is it the change associated with a one unit of BCS or 0.5 at which it was measured?

Author Response

Response to Reviewer 3 Comments

We thank the reviewer for all their comments. We have reviewed the reviewer’s comments. Each comment (point) is below in black followed by our response in red. In addition, we have made corrections where necessary and given more detail to improve the clarity of the manuscript.

Point 1:  The study uses an extensive data set and model development to relate live weight to body condition score changes. My two major concerns with the study are:

The study uses an extensive data set and model development to relate live weight to body condition score changes. My two major concerns with the study are (1) the limited effective range in body condition scores despite the large overall data set, and (2) its biological relevance. Of the 128,753 ewe records, over 97% represented only 3 BCS scores (2.5, 3.0 and 3.5) on the 9 point scale (just the two scores 2.5 and 3.0 represent more than 85% of records). While I appreciate the constraints of identifying a wider range of scores in a commercial setting and not being a statistician, I am sure how a meaningful regression analysis can be conducted over such a narrow range. This would also explain why such low r-values were obtained from such a large data set.

As to the second concern, is there is really a need to predict BCS from live weight (line 377) or use live weight as a proxy for BCS (line 380/381) when BSC is a measure that can be more readily recorded than live weight? Not sure this represents valid conclusions. A stronger case needs to be made by the authors as to the need for these prediction models and their practical applications.

Response to point 1:

In regard to the limited effect range. The magnitude of the correlation coefficients is known to be influenced by several factors [1] such as: i) variability in the data, ii) distributions and shapes of x and y measures, iii) scale range, iv) presence or absence of a linear relationship, v) how reliable are x and y measures, vi) presence of influential outliers. Our data was rigorously subjected to all the above criterion and was found to be fit for the analysis used in the methods section.

For instance, as shown in the methods and results;

  • concerning the reliability of the measurements, all BCS measurements were taken by experienced operators (see lines 129 to 130, page 3).
  • variability was within comparable ranges for most of the BCS data points, with CV < 20% (see Table 2, page 6),
  • the results indicated a linear association between LW and BCS (see lines 340 to 343, page 10),
  • the scatterplot showed no obvious influential outliers (page 6) and in addition, we indicated that those found during the final analysis to be so were removed (see lines 169 to 170, page 4),
  • the distribution shapes of the LW and BCS measurements using the 1-5 BCS scale were similar (results not shown) and
  • about the scale range, the correlation coefficients for the relationship between LW and BCS were still comparable for the full (1-5) and narrow (2.5-3.5) ranges of BCS. Moreover, the observed correlation was moderate but comparable to findings from other studies which used narrow ranges (see lines 384 to 385, page 11).

In the analysis we used linear regression which unlike classification related analysis, circumvents biases associated with class imbalances (especially too few counts some categories than others) making it an appropriate for analysis over different ranges of values on continuous and discrete variables [2-4]. Moreover, the observed correlation was moderate but comparable to findings from other studies which used narrow ranges. Certainly, we agree that we would get more accurate results with more data from a wider range of values. Therefore, we believe in relation to this point no change is required.

In regard to biological significance and need to predict body condition score from live weight. Body condition score has a significant impact on sheep performance (see review [5]).  As body condition drops below the optimal range of 3.0 to 3.5 animal performance decreases significantly and above the optimum range there is no further improvement in performance and thus efficiency decreases due to greater nutritional cost. Therefore, it is important that farmers can identify those animals that need to gain condition, maintain condition and, if required, those that can lose condition. Body condition scoring is promoted as a management tool for optimal nutritional manage of a flock (see lines 45 to 50). Body condition score circumvents the limitations of live weight such as variations in gut-fill (see lines 42 to 44). However, unlike BCS, live weight is not a subjective measure and can be accurately measured using weighing systems and lifetime data can be recorded with these systems. Further, there is ‘operator’ variation when measuring a subjective trait like BCS [5].  With modern weighing systems 300 to 500 sheep can be weighed per hour. On many farms (especially large southern hemisphere sheep farms with 1000 to 5000 sheep), BCS is utilized by a low percentage of farmers (see lines 54 to 59) as it is labour intensive (and on many of these farms labour units are low) and an additional procedure to weighing. If accurate prediction equations can be developed, use of BCS as a tool will be increased and improved nutritional management of ewes will occur. Therefore, we have incorporated a number of points in the introduction as a justification for predicting BCS based on live weight (see lines 51 to 69, page 2).

Point 2:  Line 24; ‘Pregnancy diagnosis’ is an arbitrary measure dependent on management system, ‘mid-pregnancy’ should be used to more clearly define stage of production cycle.

Response to point 2: changes made (page 3).

We have changed this to ‘at pregnancy diagnosis’ throughout the entire manuscript. The timing of this is already outlined in the methods as being in mid pregnancy (lines 125 to 127) and it is well established that pregnancy diagnosis in sheep with ultrasound can only occur in mid-pregnancy so there is no need to change this point.  We also wish to use this term to be clear to practitioners that BCS at the time of pregnancy diagnosis is an important timing for this tool.

Point 3: Line 28; ‘Pregnancy-rank’ should be replaced with ‘pregnancy status’ to more accurately reflect what the authors are referring to. Also, with this many parturitions, were there no triplet birth? If so, the pregnancy status should be changed to ‘multiples’, unless these records were deleted from the data set.

Response to point 3:

We think pregnancy-rank is more accurate than pregnancy status as the latter can mean being pregnant or not. In this study we use pregnancy-rank to refer to ewes that are already pregnant but want to describe them by the number of foetuses they carry.

There were some triplet cases (n=67) recorded, which, distributed by the number of years, would result in small sample sizes for meaning full comparisons. In addition, these triplets had the most variability (both LW and BCS) among the few triplet cases and thus, were not considered during the analysis. We have incorporated in a statement about the triplets (see lines 146 to 147, page 3)

Point 4: Line 77; electronic ID and modern weigh system would not be ‘pre-requisites’ for a study to collect lifetime data (given it would help)

Response to point 4: changes made (see lines 97 to 99, page 3).

We have added more detail and now the sentence reads “Both conventional and modern weighing systems combined with individual electronic identification can now allow lifetime data to be collected more easily and quickly on large sheep flocks.”.

Point 5: Line 93; table A1 and A2 refer to animal numbers not management

Response to point 5: change made (see line 113, page 3).

Deleted the words table A1 and A2 and now line reads “… eight months of age at breeding.”

Point 6: Line 110; there is no table A4

Response to point 6: change made (see line 131, page 3).

The word “Table A4” corrected to “Table A2” and now line reads “… summarized in Tables A1 and A2. Additional …”

Point 7: Figures A4/5; what is the relevance of measuring significance of differences in live weight across age groups and production stages, wouldn’t it be more meaningful if it would be corrected for these variables (of course a ewe is heavier when late pregnant and then at breeding)

Response to point 7: changes made (see page 3, table 4, page 8; table 5, page 9).

We now only highlight the significance of individual regression and correlation coefficients in the tables 4 and 5 but kept them in figures A4 and A5. Also, in regard to the test associated with these tables (see lines 200 to 201). It was not possible to correct for live weight weights for exact stage of pregnancy as we did not have all required pregnancy and birth information given the extensive nature of our study (i.e. it was not possible with more than 3000 ewe lambing each year outdoors to collect individual birthing day (see lines 134 to 135).

Point 8: Line 224; superscripts in Figure A4 would suggest that live weight of single and twin bearing ewes are not different at 55-66 and >67 months as is stated in the text

Response to point 8:  changes made (see lines 256 to 258, pages 7)

We have replaced age group 55-66 months with 43-54 months for the comparison between twin and single bearing ewes during pregnancy diagnosis and replaced 43-54 with 55-66 months during pre-lambing. The lines now read “twin bearing ewes consistently had greater (P<0.05) live weight than single or non-pregnant ewes across age up to the 43-54 months. Pre-lambing, live weight was greater in twin than single bearing ewes (P < 0.01) up to the 55-66 months”.

Point 9: Line 235; BCS according to the superscripts in Figure A6 indicate that differences in BCS between age groups was not significant, also title for Figure A6 should say ‘body condition score’, rather than ‘live weight’

Response to point 9: changes made (see lines 266, page 7, 22)

The significant p value notation has been removed as it was misleading and a mistake. The line now reads “Within the annual cycle and over time, the BCS of ewes showed no clear pattern of decline.” Which agrees with Figure A6. In addition, the title for figure A6 (page 22) has been corrected and now reads “The body condition score of ewes …”

Point 10: Line 265; define incremental live weight change here, is it the change associated with a one unit of BCS or 0.5 at which it was measured?

Response to point 10: changes made (see line 290 to 291, page 8).

We have added the word ‘one’ to the definition of incremental live weight change and now reads “… average change in live weight per one-unit change in BCS (incremental liveweight change), were affected by all three factors (P < 0.05).” 

References

  1. Goodwin L.D.; Leech N.L. Understanding correlation: Factors that affect the size of r. The Journal of Experimental Education 2006, 74(3):249-66.
  2. He H.; Garcia E.A. Learning from imbalanced data. IEEE Transactions on Knowledge & Data Engineering 2008, (9):1263-84.
  3. Tharwat A. Classification assessment methods. Applied Computing and Informatics 2018.
  4. Leevy J.L.; Khoshgoftaar T.M.; Bauder R.A.; Seliya N. A survey on addressing high-class imbalance in big data. Journal of Big Data 2018, 5(1):42.
  5. Kenyon P.R.; Maloney S.K.; Blache D. Review of sheep body condition score in relation to production characteristics. N Z J Agric Res 2014, 57(1):38-64.
